# DIFFUSION MODEL-AUGMENTED BEHAVIORAL CLONING

## ABSTRACT

Imitation learning addresses the challenge of learning by observing an expert's demonstrations without access to reward signals from environments. Most existing imitation learning methods that do not require interacting with environments either model the expert distribution as the conditional probability $p(a|s)$ (*e.g.*, behavioral cloning, BC) or the joint probability $p(s, a)$ Despite its simplicity, modeling the conditional probability with BC usually struggles with generalization. While modeling the joint probability can lead to improved generalization performance, the inference procedure is often time-consuming and the model can suffer from manifold overfitting. This work proposes an imitation learning framework that benefits from modeling both the conditional and joint probability of the expert distribution. Our proposed diffusion model-augmented behavioral cloning (DBC) employs a diffusion model trained to model expert behaviors and learns a policy to optimize both the BC loss (conditional) and our proposed diffusion model loss (joint). DBC outperforms baselines in various continuous control tasks in navigation, robot arm manipulation, dexterous manipulation, and locomotion. We design additional experiments to verify the limitations of modeling either the conditional probability or the joint probability of the expert distribution as well as compare different generative models. Ablation studies justify the effectiveness of our design choices.

## 1 INTRODUCTION

Recently, the success of deep reinforcement learning (DRL) (Mnih et al., 2015; Lillicrap et al., 2016; Arulkumaran et al., 2017) has inspired the research community to develop DRL frameworks to control robots, aiming to automate the process of designing sensing, planning, and control algorithms by letting the robot learn in an end-to-end fashion. Yet, acquiring complex skills through trial and error can still lead to undesired behaviors even with sophisticated reward design (Christiano et al., 2017; Leike et al., 2018; Lee et al., 2019). Moreover, the exploring process could damage expensive robotic platforms or even be dangerous to humans (Garcıa and Fernández, 2015; Levine et al., 2020).

To overcome this issue, imitation learning (*i.e.*, learning from demonstration) (Schaal, 1997; Osa et al., 2018) has received growing attention, whose aim is to learn a policy from expert demonstrations, which are often more accessible than appropriate reward functions for reinforcement learning. Among various imitation learning directions, adversarial imitation learning (Ho and Ermon, 2016; Zolna et al., 2021; Kostrikov et al., 2019) and inverse reinforcement learning (Ng and Russell, 2000; Abbeel and Ng, 2004) have achieved encouraging results in a variety of domains. Yet, these methods require interacting with environments, which can still be expensive or even dangerous.

On the other hand, behavioral cloning (BC) (Pomerleau, 1989; Bain and Sammut, 1995) does not require interacting with environments. BC formulates imitation learning as a supervised learning problem — given an expert demonstration dataset, an agent policy takes states sampled from the dataset as input and learns to replicate the corresponding expert actions. One can view a BC policy as a discriminative model $p(a|s)$ that models the *conditional probability* of actions $a$ given a state $s$. Due to its simplicity and training stability, BC has been widely adopted for various applications. However, BC struggles at generalizing to states unobserved during training (Nguyen et al., 2023).

To alleviate the generalization issue, we propose to augment BC by modeling the *joint probability* $p(s, a)$ of expert state-action pairs with a generative model (*e.g.*, diffusion models). This is motivated

by Bishop and Nasrabadi (2006) and Fisch et al. (2013), who illustrate that modeling joint probability allows for better generalizing to data points unobserved during training. However, with a learned joint probability model $p(s, a)$, retrieving a desired action $a$ requires actions sampling and optimization (*i.e.*, $\arg\max_{a \in \mathcal{A}} p(s, a)$), which can be extremely inefficient with a large action space. Moreover, modeling joint probabilities can suffer from manifold overfitting (Wu et al., 2021; Loaiza-Ganem et al., 2022) when observed high-dimensional data lies on a low-dimensional manifold (*e.g.*, state-action pairs collected from a script expert policies).

This work proposes an imitation learning framework that combines both the efficiency and stability of modeling the *conditional probability* and the generalization ability of modeling the *joint probability*. Specifically, we propose to model the expert state-action pairs using a state-of-the-art generative model, a diffusion model, which learns to estimate how likely a state-action pair is sampled from the expert dataset. Then, we train a policy to optimize both the BC objective and the estimate produced by the learned diffusion model. Therefore, our proposed framework not only can efficiently predict actions given states via capturing the *conditional probability* $p(a|s)$ but also enjoys the generalization ability induced by modeling the *joint probability* $p(s, a)$ and utilizing it to guide policy learning.

We evaluate our proposed framework and baselines in various continuous control domains, including navigation, robot arm manipulation, and locomotion. The experimental results show that the proposed framework outperforms all the baselines or achieves competitive performance on all tasks. Extensive ablation studies compare our proposed method to its variants, justifying our design choices, such as different generative models, and investigating the effect of hyperparameters.

## 2 RELATED WORK

Imitation learning addresses the challenge of learning by observing expert demonstrations without access to reward signals from environments. It has various applications such as robotics (Schaal, 1997; Zhao et al., 2023), autonomous driving (Ly and Akhloufi, 2020), and game AI (Harmer et al., 2018).

**Behavioral Cloning (BC).** BC (Pomerleau, 1989; Torabi et al., 2018) formulates imitating an expert as a supervised learning problem. Due to its simplicity and effectiveness, it has been widely adopted in various domains. Yet, it often struggles at generalizing to states unobserved from the expert demonstrations (Ross et al., 2011; Florence et al., 2022). In this work, we augment BC by employing a diffusion model that learns to capture the joint probability of expert state-action pairs.

**Adversarial Imitation Learning (AIL).** AIL methods aim to match the state-action distributions of an agent and an expert via adversarial training. Generative adversarial imitation learning (GAIL) (Ho and Ermon, 2016) and its extensions (Torabi et al., 2019; Kostrikov et al., 2019; Zolna et al., 2021) resemble the idea of generative adversarial networks (Goodfellow et al., 2014), which trains a generator policy to imitate expert behaviors and a discriminator to distinguish between the expert and the learner's state-action pair distributions. While modeling state-action distributions often leads to satisfactory performance, adversarial learning can be unstable and inefficient (Chen et al., 2020). Moreover, AIL methods require online interaction with environments, which can be costly or even dangerous. In contrast, our work does not require interacting with environments.

**Inverse Reinforcement Learning (IRL).** IRL methods (Ng and Russell, 2000; Abbeel and Ng, 2004; Fu et al., 2018; Lee et al., 2021) are designed to infer the reward function that underlies the expert demonstrations and then learn a policy using the inferred reward function. This allows for learning tasks whose reward functions are difficult to specify manually. However, due to its double-loop learning procedure, IRL methods are typically computationally expensive and time-consuming. Additionally, obtaining accurate estimates of the expert's reward function can be difficult, especially when the expert's behavior is non-deterministic or when the expert's demonstrations are sub-optimal.

**Diffusion Policies.** Recently, Pearce et al. (2023); Chi et al. (2023); Reuss et al. (2023) propose to represent and learn an imitation learning policy using a conditional diffusion model, which produces a predicted action conditioning on a state and a sampled noise vector. These methods achieve encouraging results in modeling stochastic and multimodal behaviors from human experts or play data. In contrast, instead of representing a policy using a diffusion model, our work employs a diffusion model trained on expert demonstrations to guide a policy as a learning objective.

## 3 Preliminaries

### 3.1 Imitation Learning

In contrast to reinforcement learning, whose goal is to learn a policy $\pi$ based on rewards received while interacting with the environment, imitation learning methods aim to learn the policy from an expert demonstration dataset containing $M$ trajectories, $D = \{\tau_1, ..., \tau_M\}$, where $\tau_i$ represents a sequence of $n_i$ state-action pairs $\{s_1^i, a_1^i, ..., s_{n_i}^i, a_{n_i}^i\}$.

#### 3.1.1 Modeling Conditional Probability $p(a|s)$

To learn a policy $\pi$, behavioral cloning (BC) directly estimates the expert policy $\pi^E$ with maximum likelihood estimation (MLE). Given a state-action pair $(s, a)$ sampled from the dataset $D$, BC optimizes $\max_\theta \sum_{(s,a) \in D} \log(\pi_\theta(a|s))$, where $\theta$ denotes the parameters of the policy $\pi$. One can view a BC policy as a discriminative model $p(a|s)$, capturing the *conditional probability* of an action $a$ given a state $s$. Despite its success in various applications, BC tends to overfit and struggle at generalizing to states unseen during training (Ross et al., 2011; Codevilla et al., 2019; Wang et al., 2022).

#### 3.1.2 Modeling Joint Probability $p(s, a)$

On the other hand, modeling the *joint probability* can yield improved generalization performance, as illustrated in Bishop and Nasrabadi (2006); Fisch et al. (2013). For instance, Florence et al. (2022); Ganapathi et al. (2022) propose to model the *joint probability* $p(s, a)$ of expert state-action pairs using an energy-based model. Then, during inference, a gradient-free optimizer is used to retrieve a desired action $a$ by sampling and optimizing actions (*i.e.*, $\arg \max_{a \in \mathcal{A}} p(s, a)$). Despite its success in various domains, it can be extremely inefficient to retrieve actions with a large action space.

Moreover, explicit generative models such as energy-based models (Du and Mordatch, 2019; Song and Kingma, 2021), variational autoencoder (Kingma and Welling, 2014), and flow-based models (Rezende and Mohamed, 2015; Dinh et al., 2017) are known to struggle with modeling observed high-dimensional data that lies on a low-dimensional manifold (*i.e.*, manifold overfitting) (Wu et al., 2021; Loaiza-Ganem et al., 2022). As a result, these methods often perform poorly when learning from demonstrations produced by script policies or PID controllers, as discussed in Section 5.4.

We aim to develop an imitation learning framework that enjoys the advantages of modeling the *conditional probability* $p(a|s)$ and the *joint probability* $p(s, a)$. Specifically, we propose to model the *joint probability* of expert state-action pairs using an explicit generative model $\phi$, which learns to produce an estimate indicating how likely a state-action pair is sampled from the expert dataset. Then, we train a policy to model the *conditional probability* $p(a|s)$ by optimizing the BC objective and the estimate produced by the learned generative model $\phi$. Hence, our method can efficiently predict actions given states, generalize better to unseen states, and suffer less from manifold overfitting.

### 3.2 Diffusion Models

As described in the previous sections, this work aims to combine the advantages of modeling the *conditional probability* $p(a|s)$ and the *joint probability* $p(s, a)$. To this end, we leverage diffusion models to model the *joint probability* of expert state-action pairs. The diffusion model is a recently developed class of generative models and has achieved state-of-the-art performance on various tasks (Sohl-Dickstein et al., 2015; Nichol and Dhariwal, 2021; Dhariwal and Nichol, 2021).

In this work, we utilize Denoising Diffusion Probabilistic Models (DDPMs) (J Ho, 2020) to model expert state-action pairs. Specifically, DDPM models gradually add noise to data samples (*i.e.*, concatenated state-action pairs) until they become isotropic Gaussian (*forward diffusion process*), and then learn to denoise each step and restore the original data samples (*reverse diffusion process*), as illustrated in Figure 1. In other words, DDPM learns to recognize a data distribution by learning to denoise noisy sampled data. More discussion on diffusion models can be found in the Section J.

## 4 Approach

Our goal is to design an imitation learning framework that enjoys both the advantages of modeling the *conditional probability* and the *joint probability* of expert behaviors. To this end, we first adopt behavioral cloning (BC) for modeling the *conditional probability* from expert state-action pairs, as described in Section 4.1. To capture the *joint probability* of expert state-action pairs, we employ a diffusion model which learns to produce an estimate indicating how likely a state-action pair is sampled from the expert state-action pair distribution, as presented in Section 4.2.1. Then, we propose to guide the policy learning by optimizing this estimate provided by

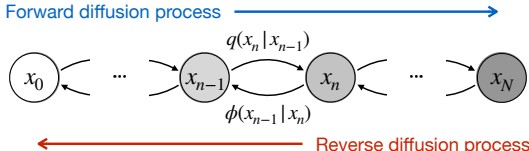

Figure 1: **Denoising Diffusion Probabilistic Model (DDPM).** Latent variables $x_1, ..., x_N$ are produced from the data point $x_0$ via the forward diffusion process, *i.e.*, gradually adding noises to the latent variables. The diffusion model $\phi$ learns to reverse the diffusion process by denoising the noisy data to reconstruct the original data point $x_0$.

a learned diffusion model, encouraging the policy to produce actions similar to expert actions, as discussed in Section 4.2.2. Finally, in Section 4.3, we introduce the framework that combines the BC loss and our proposed diffusion model loss, allowing for learning a policy that benefits from modeling both the *conditional probability* and the *joint probability* of expert behaviors. An overview of our proposed framework is illustrated in Figure 2, and the algorithm is detailed in Section A.

## 4.1 BEHAVIORAL CLONING LOSS

The behavioral cloning (BC) model aims to imitate expert behaviors with supervision learning. BC learns to capture the conditional probability $p(a|s)$ of expert state-action pairs. A BC policy $\pi(a|s)$ learns by optimizing

$$\mathcal{L}_{\text{BC}} = \mathbb{E}_{(s,a)\sim D, \hat{a}\sim\pi(s)}[d(a, \hat{a})],\tag{1}$$

where $d(\cdot, \cdot)$ denotes a distance measure between a pair of actions. For example, we can adapt the mean-square error (MSE) loss $||a - \hat{a}||^2$ for most continuous control tasks.

## 4.2 LEARNING A DIFFUSION MODEL AND GUIDING POLICY LEARNING

Instead of directly learning the conditional probability $p(a|s)$, this section discusses how to model the joint probability $p(s, a)$ of expert behaviors with a diffusion model in Section 4.2.1 and presents how to leverage the learned diffusion model to guide policy learning in Section 4.2.2.

### 4.2.1 LEARNING A DIFFUSION MODEL

We propose to model the joint probability of expert state-action pairs with a diffusion model $\phi$. Specifically, we create a joint distribution by simply concatenating a state vector $s$ and an action vector $a$ from a state-action pair $(s, a)$. To model such distribution by learning a denoising diffusion probabilistic model (DDPM) (J Ho, 2020), we inject noise $\epsilon(n)$ into sampled state-action pairs, where $n$ indicates the number of steps of the Markov procedure, which can be viewed as a variable of the level of noise, and the total number of steps is notated as $N$. Then, we train the diffusion model $\phi$ to predict the injected noises by optimizing

$$\mathcal{L}_{\text{diff}}(s, a, \phi) = \mathbb{E}_{n\sim N, (s,a)\sim D}[||\hat{\epsilon}(s, a, n) - \epsilon(n)||^2] = \mathbb{E}_{n\sim N, (s,a)\sim D}[||\phi(s, a, \epsilon(n)) - \epsilon(n)||^2],\tag{2}$$

where $\hat{\epsilon}$ is the noise predicted by the diffusion model $\phi$. Once optimized, the diffusion model can *recognize* the expert distribution by perfectly predicting the noise injected into state-action pairs sampled from the expert distribution. On the other hand, predicting the noise injected into state-action pairs sampled from any other distribution should yield a higher loss value. Therefore, we propose to view $\mathcal{L}_{\text{diff}}(s, a, \phi)$ as an estimate of how well the state-action pair $(s, a)$ fits the state-action distribution that $\phi$ learns from.

### 4.2.2 LEARNING A POLICY WITH DIFFUSION MODEL LOSS

A diffusion model $\phi$ trained on an expert dataset can produce an estimate $\mathcal{L}_{\text{diff}}(s, a, \phi)$ indicating how well a state-action pair $(s, a)$ fits the expert distribution. We propose to leverage this signal to guide a

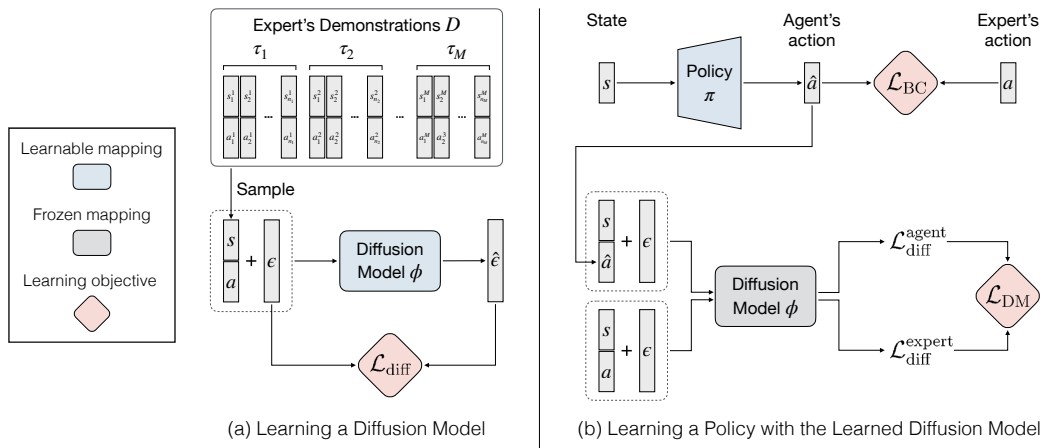

Figure 2: **Diffusion Model-Augmented Behavioral Cloning.** Our proposed method DBC augments behavioral cloning (BC) by employing a diffusion model. (a) **Learning a Diffusion Model**: the diffusion model $\phi$ learns to model the distribution of concatenated state-action pairs sampled from the demonstration dataset $D$. It learns to reverse the diffusion process (*i.e.*, denoise) by optimizing $\mathcal{L}_{\text{diff}}$ in Eq. 2. (b) **Learning a Policy with the Learned Diffusion Model**: we propose a diffusion model objective $\mathcal{L}_{\text{DM}}$ for policy learning and jointly optimize it with the BC objective $\mathcal{L}_{\text{BC}}$. Specifically, $\mathcal{L}_{\text{DM}}$ is computed based on processing a sampled state-action pair $(s, a)$ and a state-action pair $(s, \hat{a})$ with the action $\hat{a}$ predicted by the policy $\pi$ with $\mathcal{L}_{\text{diff}}$.

policy $\pi$ predicting actions $\hat{a}$ to imitate the expert. Specifically, the policy $\pi$ learns by optimizing

$$\mathcal{L}_{\text{diff}}^{\text{agent}} = \mathcal{L}_{\text{diff}}(s, \hat{a}, \phi) = \mathbb{E}_{s \sim D, \hat{a} \sim \pi(s)}[||\hat{\epsilon}(s, \hat{a}, n) - \epsilon||^2]. \tag{3}$$

Intuitively, the policy $\pi$ learns to predict actions $\hat{a}$ that are indistinguishable from the expert actions $a$ for the diffusion model conditioning on the same set of states.

We hypothesize that learning a policy to optimize Eq. 3 can be unstable, especially for state-action pairs that are not well-modeled by the diffusion model, which yield a high value of $\mathcal{L}_{\text{diff}}$ even with expert state-action pairs. Therefore, we propose to normalize the agent diffusion loss $\mathcal{L}_{\text{diff}}^{\text{agent}}$ with an expert diffusion loss $\mathcal{L}_{\text{diff}}^{\text{expert}}$, which can be computed with expert state-action pairs $(s, a)$ as follows:

$$\mathcal{L}_{\text{diff}}^{\text{expert}} = \mathcal{L}_{\text{diff}}(s, a, \phi) = \mathbb{E}_{(s,a) \sim D}[||\hat{\epsilon}(s, a, n) - \epsilon||^2]. \tag{4}$$

We propose to optimize the diffusion model loss $\mathcal{L}_{\text{DM}}$ based on calculating the difference between the above agent and expert diffusion losses:

$$\mathcal{L}_{\text{DM}} = \mathbb{E}_{(s,a) \sim D, \hat{a} \sim \pi(s)}[max(\mathcal{L}_{\text{diff}}^{\text{agent}} - \mathcal{L}_{\text{diff}}^{\text{expert}}, 0)]. \tag{5}$$

### 4.3 Combining the Two Objectives

Our goal is to learn a policy that benefits from both modeling the conditional probability and the joint probability of expert behaviors. To this end, we propose to augment a BC policy that optimizes the BC loss $L_{\text{BC}}$ in Eq. 1 by jointly optimizing the proposed diffusion model loss $L_{\text{DM}}$ in Eq. 5, which encourages the policy to predict actions that fit the expert joint probability captured by a diffusion model. To learn from both the BC loss and the diffusion model loss, we train the policy to optimize

$$\mathcal{L}_{\text{total}} = \mathcal{L}_{\text{BC}} + \lambda \mathcal{L}_{\text{DM}}, \tag{6}$$

where $\lambda$ is a coefficient that determines the importance of the diffusion model loss relative to the BC loss. Our experimental results empirically show that optimizing a combination of the BC loss $\mathcal{L}_{\text{BC}}$ and the diffusion model loss $\mathcal{L}_{\text{DM}}$ leads to the best performance compared to solely optimizing each of them, highlighting the effectiveness of the proposed combined loss $\mathcal{L}_{\text{total}}$. Further discussions on combing these two losses can be found in Section B.

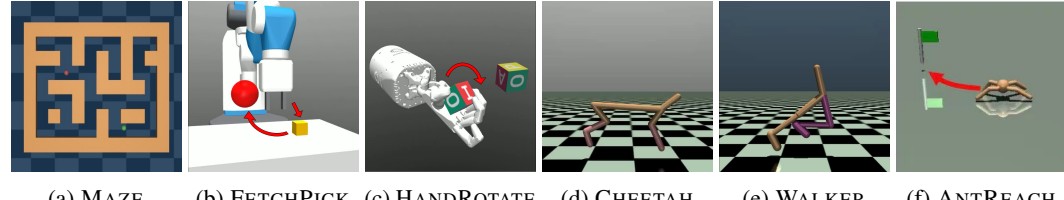

| (a) MAZE | (b) FETCHPICK | (c) HANDROTATE | (d) CHEETAH | (e) WALKER | (f) ANTREACH |

Figure 3: **Environments & Tasks. (a) MAZE**: A point-mass agent (green) in a 2D maze learns to navigate from its start location to a goal location (red). **(b) FETCHPICK**: The robot arm manipulation tasks employ a 7-DoF Fetch robotics arm to pick up an object (yellow cube) from the table and move it to a target location (red). **(c) HANDROTATE**: This dexterous manipulation task requires a Shadow Dexterous Hand to in-hand rotate a block to a target orientation. **(d)-(e) CHEETAH and WALKER**: These locomotion tasks require learning a 2d-dimensional agent to walk as fast as possible while maintaining its balance. **(f) ANTREACH**: This task combines locomotion and navigation, instructing an ant to reach the goal while maintaining its balance.

## 5 EXPERIMENTS

We design experiments in various continuous control domains, including navigation, robot arm manipulation, dexterous manipulation, and locomotion, to compare our proposed framework (DBC) to its variants and baselines.

### 5.1 EXPERIMENTAL SETUP

This section describes the environments, tasks, and expert demonstrations used for learning and evaluation. More details can be found in Section E.

**Navigation.** To evaluate our method on a navigation task, we choose MAZE, a maze environment proposed in (Fu et al., 2020) (maze2d-medium-v2), as illustrated in Figure 3a. This task features a point-mass agent in a 2D maze learning to navigate from its start location to a goal location by iteratively predicting its $x$ and $y$ acceleration. The agent's beginning and final locations are chosen randomly. We collect 100 demonstrations with 18,525 transitions using a controller.

**Robot Arm Manipulation.** We evaluate our method in FETCHPICK, a robot arm manipulation domain with a 7-DoF Fetch task, as illustrated in Figure 3b. FETCHPICK requires picking up an object from the table and lifting it to a target location. We use the demonstrations, consisting of 10k transitions (303 trajectories), provided by Lee et al. (2021) for these tasks.

**Dexterous Manipulation.** In HANDROTATE, we further evaluate our method on a challenging environment proposed in Plappert et al. (2018), where a 24-DoF Shadow Dexterous Hand learns to in-hand rotate a block to a target orientation, as illustrated in Figure 3c. This environment has a state space (68D) and action space (20D), which is high dimensional compared to the commonly-used environments in IL. We collected 10k transitions (515 trajectories) from a SAC (Haarnoja et al., 2018) expert policy trained for 10M environment steps.

**Locomotion.** For locomotion, we leverage the CHEETAH and WALKER (Brockman et al., 2016) environments. Both CHEETAH and WALKER require a bipedal agent (with different structures) to travel as fast as possible while maintaining its balance, as illustrated in Figure 3d and Figure 3e, respectively. We use the demonstrations provided by Kostrikov (2018), which contains 5 trajectories with 5k state-action pairs for both the CHEETAH and WALKER environments.

**Locomotion + Navigation.** We further explore our method on the challenging ANTREACH environment. In the environment, the quadruped ant aims to reach a randomly generated target located along the boundary of a semicircle centered around the ant, as illustrated in Figure 3f. ANTREACH environment combines the properties of locomotion and goal-directed navigation tasks, which require robot controlling and path planning to reach the goal. We use the demonstrations provided by Lee et al. (2021), which contains 500 trajectories with 25k state-action pairs in ANTREACH.

Table 1: **Experimental Result.** We report the mean and the standard deviation of success rate (MAZE, FETCHPICK, HANDROTATE, ANTREACH) and return (CHEETAH, WALKER), evaluated over three random seeds. Our proposed method (DBC) outperforms or performs competitively against the best baseline over all environments.

| Method | MAZE | FETCHPICK | HANDROTATE | CHEETAH | WALKER | ANTREACH |
|---|---|---|---|---|---|---|
| BC | 92.1% ± 3.6% | 91.6% ± 5.8% | 57.5% ± 4.7% | **4873.3** ± 69.7 | 6954.4 ± 73.5 | **73.6**% ± 2.9% |
| Implicit BC | 78.3% ± 6.0% | 69.4% ± 7.3% | 13.8% ± 3.7% | 1563.6 ± 486.8 | 839.8 ± 104.2 | 34.5% ± 5.4% |
| Diffusion Policy | **95.5**% ± 1.9% | 93.9% ± 3.4% | **61.7**% ± 4.3% | 4650.3 ± 59.9 | 6479.1 ± 238.6 | 64.5% ± 3.4% |
| DBC (Ours) | **95.4**% ± 1.7% | **96.9**% ± 1.7% | **60.1**% ± 4.4% | **4909.5** ± 73.0 | **7034.6** ± 33.7 | **75.5**% ± 3.5% |

## 5.2 BASELINES

This work focuses on imitation learning problem *without* environment interactions. Therefore, approaches that require environmental interactions, such as GAIL-based methods, are not applicable. Instead, we extensively compared our proposed method to state-of-the-art imitation learning methods that do not require interaction with the environment, including Implicit BC (Florence et al., 2022) and Diffusion Policy (Chi et al., 2023; Reuss et al., 2023).

- **BC** learns to imitate an expert by modeling the conditional probability $p(a|s)$ of the expert behaviors via optimizing the BC loss $\mathcal{L}_{BC}$ in Eq. 1.

- **Implicit BC (IBC)** Florence et al. (2022) models expert state-action pairs with an energy-based model. For inference, we implement the derivative-free optimization algorithm proposed in IBC, which samples actions iteratively and select the desired action according to the predicted energies.

- **Diffusion policy** refers to the methods that learn a conditional diffusion model as a policy (Chi et al., 2023; Reuss et al., 2023). Specifically, we implement this baseline based on Pearce et al. (2023). We include this baseline to analyze the effectiveness of using diffusion models as a policy or as a learning objective (ours).

## 5.3 EXPERIMENTAL RESULTS

We report the experimental results in terms of success rate (MAZE, FETCHPICK, HANDROTATE, and ANTREACH), and return (CHEETAH and WALKER) in Table 1. The details of model architecture can be found in Section F. Training and evaluation details can be found in Section G. Additional analysis and experimental results can be found in Section H and Section I.

**Overall Task Performance.** In navigation (MAZE) and manipulation (FETCHPICK and HANDROTATE) tasks, our DBC performs competitively against the Diffusion Policy and outperforms the other baselines. We hypothesize that these tasks require the agent to learn from demonstrations with various behaviors. Diffusion policy has shown promising performance for capturing multi-modality distribution, while our DBC can also generalize well with the guidance of the diffusion models, so both methods achieve satisfactory results.

On the other hand, in tasks that locomotion is involved, i.e., CHEETAH, WALKER, and ANTREACH, our DBC outperforms Diffusion Policy and performs competitively against the simple BC baseline. We hypothesize that this is because locomotion tasks with sufficient expert demonstrations and little randomness do not require generalization during inference. The agent can simply follow the closed-loop progress of the expert demonstrations, resulting in both BC and DBC performing similarly to the expert demonstrations. On the other hand, the Diffusion Policy performs slightly worse due to its design for modeling multimodal behaviors, which is contradictory to learning from single-mode simulated locomotion tasks.

**Action Space Dimension.** The Implicit BC baseline requires time-consuming action sampling and optimization during inference, and such a procedure may not scale well to high-dimensional action spaces. Our Implicit BC baseline with a derivative-free optimizer struggles in HANDROTATE and WALKER environments, whose action dimensions are 20 and 6, respectively. This is consistent with Florence et al. (2022), which reports that the optimizer failed to solve tasks with an action dimension larger than 5. In contrast, our proposed DBC can handle high-dimensional action spaces.

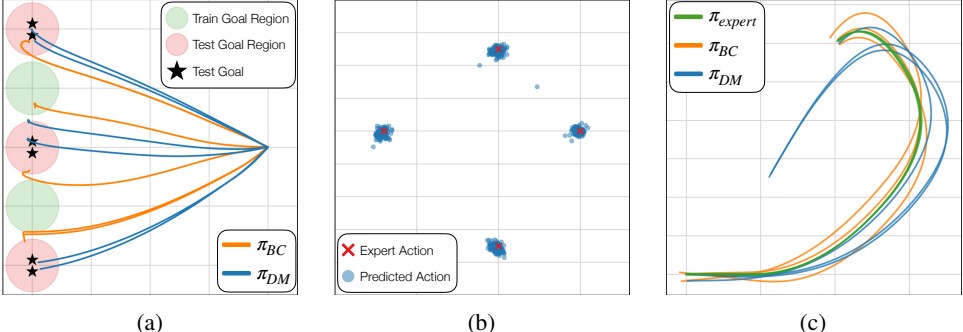

(a)    (b)    (c)

Figure 4: **Comparing Modeling Conditional Probability and Joint Probability. (a) Generalization.** We collect expert trajectories from a PPO policy learning to navigate to goals sampled from the green regions. Then, we learn a policy $\pi_{BC}$ to optimize $\mathcal{L}_{BC}$, and another policy $\pi_{DM}$ to optimize $\mathcal{L}_{DM}$ with a diffusion model trained on the expert distribution. We evaluate the two policies by sampling goals from the red regions, which requires the ability to generalize. $\pi_{BC}$ (orange) struggles at generalizing to unseen goals, whereas $\pi_{DM}$ (blue) can generalize (*i.e.*, extrapolate) to some extent. **(b)-(c) Manifold overfitting.** We collect the green spiral trajectories from a script policy, whose actions are visualized as red crosses. We then train and evaluate $\pi_{BC}$ and $\pi_{DM}$. The trajectories of $\pi_{BC}$ (orange) can closely follow the expert trajectories (green), while the trajectories of $\pi_{DM}$ (blue) deviates from expert's. This is because the diffusion model struggles at modeling such expert action distribution with a lower intrinsic dimension, which can be observed from incorrectly predicted actions (blue dots) produced by the diffusion model.

**Inference Efficiency.** To evaluate the inference efficiency, we measure and report the number of evaluation episodes per second ($\uparrow$) for Implicit BC (9.92), Diffusion Policy (1.38), and DBC (**30.79**) on an NVIDIA RTX 3080 Ti GPU in MAZE. As a results of modeling the conditional probability $p(a|s)$, DBC and BC can directly map states to actions during inference. In contrast, Implicit BC samples and optimizes actions, while Diffusion Policy iteratively denoises sampled noises, which are both time-consuming. This verifies the efficiency of modeling the conditional probability.

### 5.4 COMPARING MODELING CONDITIONAL PROBABILITY AND JOINT PROBABILITY

This section aims to empirically identify the limitations of modeling *either* the conditional *or* the joint probability in an open maze environment implemented with Fu et al. (2020).

**Generalization.** We aim to investigate if learning from the BC loss alone struggles at generalization (*conditional*) and examine if guiding the policy using the diffusion model loss yields improved generalization ability (*joint*). We collect trajectories of a PPO policy learning to navigate from $(5, 3)$ to goals sampled around $(1, 2)$ and $(1, 4)$ (green), as shown in Figure 4a. Given these expert trajectories, we learn a policy $\pi_{BC}$ to optimize Eq. 1 and another policy $\pi_{DM}$ to optimize Eq. 5. Then, we evaluate the two policies by sampling goals around $(1, 1)$, $(1, 3)$, and $(1, 5)$ (red), which requires the ability to generalize. Visualized trajectories of the two policies in Figure 4a show that $\pi_{BC}$ (orange) fails to generalize to unseen goals, whereas $\pi_{DM}$ (blue) can generalize (*i.e.*, extrapolate) to some extent. This verifies our motivation to augment BC with the diffusion model loss.

**Manifold overfitting.** We aim to examine if modeling the joint probability is difficult when observed high-dimensional data lies on a low-dimensional manifold (*i.e.*, manifold overfitting). We collect trajectories from a script policy that executes actions $(0.5, 0)$, $(0, 0.5)$, $(-0.7, 0)$, and $(0, -0.7)$ (red crosses in Figure 4b), each for 40 consecutive time steps, resulting the green spiral trajectories visualized in Figure 4c.

Given these expert demonstrations, we learn a policy $\pi_{BC}$ to optimize Eq. 1, and another policy $\pi_{DM}$ to optimize Eq. 5 with a diffusion model trained on the expert distribution. Figure 4b shows that the diffusion model struggles at modeling such expert action distribution with a lower intrinsic dimension. As a result, Figure 4c show that the trajectories of $\pi_{DM}$ (blue) deviates from the expert trajectories (green) as the diffusion model cannot provide effective loss. On the other hand, the trajectories of $\pi_{BC}$ (orange) are able to closely follow the expert's and result in a superior success

Table 2: **Generative Models.** We compare using different generative models to model the expert distribution and guide policy learning in MAZE.

| Method | without BC | with BC |
|--------|-----------|---------|
| BC | N/A | $92.1\% \pm 3.6\%$ |
| EBM | $39.6\% \pm 9.6\%$ | $83.3\% \pm 3.2\%$ |
| VAE | $53.1\% \pm 8.7\%$ | $90.9\% \pm 3.4\%$ |
| GAN | $54.4\% \pm 4.5\%$ | $89.6\% \pm 3.4\%$ |
| DM | $\mathbf{79.6}\% \pm 9.6\%$ | $\mathbf{95.4}\% \pm 1.7\%$ |

Table 3: **Effect of** $\lambda$**.** We experiment with different values of $\lambda$ in MAZE, each evaluated over three random seeds.

| $\lambda$ | Success Rate |
|-----------|--------------|
| 1 | $94.03\% \pm 2.25\%$ |
| 3 | $95.00\% \pm 2.21\%$ |
| 10 | $95.05\% \pm 2.49\%$ |
| 30 | $\mathbf{95.41}\% \pm 1.74\%$ |
| 100 | $94.04\% \pm 2.59\%$ |
| 300 | $95.24\% \pm 1.75\%$ |

rate. This verifies our motivation to complement modeling the joint probability with modeling the conditional probability (*i.e.*, BC).

## 5.5 COMPARING DIFFERENT GENERATIVE MODELS

Our proposed framework employs a diffusion model (DM) to model the joint probability of expert state-action pairs and utilizes it to guide policy learning. To justify our choice, we explore using other popular generative models to replace the diffusion model in MAZE. We consider energy-based models (EBMs) (Du and Mordatch, 2019; Song and Kingma, 2021), variational autoencoder (VAEs) (Kingma and Welling, 2014), and generative adversarial networks (GANs) (Goodfellow et al., 2014). Each generative model learns to model expert state-action pairs. To guide policy learning, given a predicted state-action pair $(s, \hat{a})$ we use the estimated energy of an EBM, the reconstruction error of a VAE, and the discriminator output of a GAN to optimize a policy with or without the BC loss.

Table 2 compares using different generative models to model the expert distribution and guide policy learning. All the generative model-guide policies can be improved by adding the BC loss, justifying our motivation to complement modeling the joint probability with modeling the conditional probability. With or without the BC loss, the diffusion model-guided policy achieves the best performance compared to other generative models, verifying our choice of the generative model. Training details of learning generative models and utilizing them to guide policy learning can be found in Section G.4.

## 5.6 EFFECT OF THE DIFFUSION MODEL LOSS COEFFICIENT $\lambda$

We examine the impact of varying the coefficient of the diffusion model loss $\lambda$ in Eq. 6 in MAZE. The result presented in Table 3 shows that $\lambda = 30$ yields the best performance. A higher or lower $\lambda$ leads to worse performance, demonstrating how modeling the conditional probability ($\mathcal{L}_{\mathrm{BC}}$) and the joint probability ($\mathcal{L}_{\mathrm{DM}}$) can complement each other.

## 6 CONCLUSION

We propose an imitation learning framework that benefits from modeling both the conditional probability $p(a|s)$ and the joint probability $p(s, a)$ of the expert distribution. Our proposed diffusion model-augmented behavioral cloning (DBC) employs a diffusion model trained to model expert behaviors and learns a policy to optimize both the BC loss and our proposed diffusion model loss. Specifically, the BC loss captures the conditional probability $p(a|s)$ from expert state-action pairs, which directly guides the policy to replicate the expert's action. On the other hand, the diffusion model loss models the joint distribution of expert state-action pairs $p(s, a)$, which provides an evaluation of how well the predicted action aligned with the expert distribution. DBC outperforms baselines or achieves competitive performance in various continuous control tasks in navigation, robot arm manipulation, dexterous manipulation, and locomotion. We design additional experiments to verify the limitations of modeling either the conditional probability or the joint probability of the expert distribution as well as compare different generative models. Ablation studies investigate the effect of hyperparameters and justify the effectiveness of our design choices. The limitations and the broader impacts can be found in the Appendix.

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
