APPENDIX

# Table of Contents

LIST OF TABLES

## LIST OF FIGURES

## A   DETAILED ALGORITHM

Our proposed framework DBC is detailed in Algorithm 1. The algorithm consists of two parts. (1) **Learning a diffusion model**: The diffusion model $\phi$ learns to model the distribution of concatenated state-action pairs sampled from the demonstration dataset $D$. It learns to reverse the diffusion process (*i.e.*, denoise) by optimizing $\mathcal{L}_{\text{diff}}$. (2) **Learning a policy with the learned diffusion model**: We propose a diffusion model objective $\mathcal{L}_{\text{DM}}$ for policy learning and jointly optimize it with the BC objective $\mathcal{L}_{\text{BC}}$. Specifically, $\mathcal{L}_{\text{DM}}$ is computed based on processing a sampled state-action pair $(s, a)$ and a state-action pair $(s, \hat{a})$ with the action $\hat{a}$ predicted by the policy $\pi$ with $\mathcal{L}_{\text{diff}}$.

---

**Algorithm 1** Diffusion Model-Augmented Behavioral Cloning (DBC)

---

**Input**: Expert's Demonstration Dataset $D$
**Output**: Policy $\pi$.

1:  // Learning a diffusion model $\phi$
2:  Randomly initialize a diffusion model $\phi$
3:  **for** each diffusion model iteration **do**
4:     Sample $(s, a)$ from $D$
5:     Sample noise level $n$ from $\{0, ..., N\}$
6:     Update $\phi$ using $L_{\text{diff}}$ from Eq. 2
7:  **end for**
8:  // Learning a policy $\pi$ with the learned diffusion model $\phi$
9:  Randomly initialize a policy $\pi$
10: **for** each policy iteration **do**
11:     Sample $(s, a)$ from $D$
12:     Predict an action $\hat{a}$ using $\pi$ from $s$: $\hat{a} \sim \pi(s)$
13:     Compute the BC loss $L_{\text{BC}}$ using Eq. 1
14:     Sample noise level $n$ from $\{0, ..., N\}$
15:     Compute the agent diffusion loss $L_{\text{diff}}^{\text{agent}}$ with $(s, \hat{a})$ using Eq. 3
16:     Compute the expert diffusion loss $L_{\text{diff}}^{\text{expert}}$ with $(s, a)$ using Eq. 4
17:     Compute the diffusion model loss $L_{\text{DM}}$ using Eq. 5
18:     Update $\pi$ using the total loss $L_{\text{total}}$ from Eq. 6
19: **end for**
20: **return** $\pi$

---

## B   FURTHER DISCUSSION ON COMBINING $\mathcal{L}_{\text{BC}}$ AND $\mathcal{L}_{\text{DM}}$

### B.1   THE DIFFERENCE AND THE COMPATIBILITY BETWEEN $\mathcal{L}_{\text{BC}}$ AND $\mathcal{L}_{\text{DM}}$

Since we propose to combine $\mathcal{L}_{\text{DM}}$ and $\mathcal{L}_{\text{BC}}$ as illustrated in Section 4.3, in the following paragraph, we will explain the difference between them and the compatibility of combing them. From a theoretical perspective, the joint probability $p(s, a)$, which is modeled by minimizing $\mathcal{L}_{\text{DM}}$, can be represented as the product of the marginal state probability and the conditional action probability using the Bayes Rules, i.e., $p(s, a) = p(s)p(a|s)$. In short, $\mathcal{L}_{\text{DM}}$ takes $p(s)$ into account to model the joint distribution while $\mathcal{L}_{\text{BC}}$ optimizes $p(a|s)$ directly.

Observing that despite their difference, when $\pi$ converges to $\pi^E$, both $\mathcal{L}_{\text{BC}}$ and $\mathcal{L}_{\text{DM}}$ converge to 0, indicating that these two losses are not conflicting. Moreover, our experimental results show that optimizing a combination of these two losses leads to the best performance, compared to solely optimizing each of them. Table 1 shows that DBC ($\mathcal{L}_{\text{BC}} + \mathcal{L}_{\text{DM}}$) outperforms BC ($\mathcal{L}_{\text{BC}}$) and Table 2 shows that optimizing $\mathcal{L}_{\text{BC}} + \mathcal{L}_{\text{DM}}$ outperforms solely optimizing $\mathcal{L}_{\text{DM}}$.

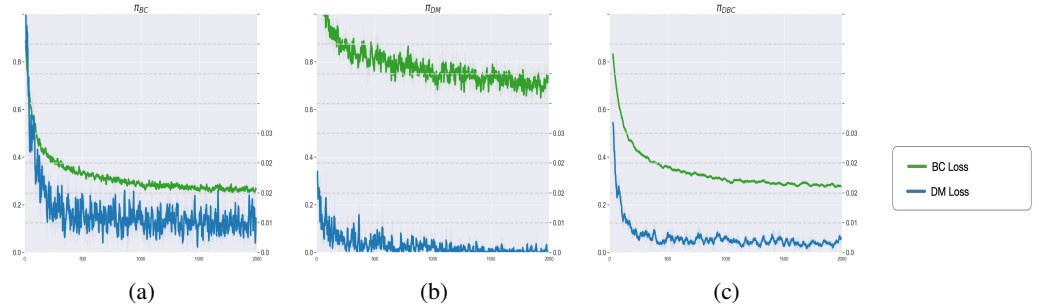

|   |   |   |
|:-:|:-:|:-:|
| (a) | (b) | (c) |

Figure 5: **Training Loss Curve.** The $\mathcal{L}_{BC}$ loss curve and $\mathcal{L}_{DM}$ loss curve of three different training conditions: **(a) update with $\mathcal{L}_{BC}$ solely**, **(b) update with $\mathcal{L}_{DM}$**, and **(c) our proposed DBC**

Table 4: Expert distribution modeling with diffusion models trained with different noise levels.

| **Noise level** | 0 | 0.002 | 0.005 | 0.01 | 0.02 | 0.05 | 0.1 |
|---|---|---|---|---|---|---|---|
| MSE Distance | 0.0213 | 0.0217 | 0.0248 | 0.0218 | 0.0235 | 0.0330 | 0.0507 |

As shown in Figure 5, even BC only optimizes $\mathcal{L}_{BC}$, $\mathcal{L}_{DM}$ also reduces. However, $\mathcal{L}_{DM}$ of BC converges to a higher value (0.0056), compared to only optimizing $\mathcal{L}_{DM}$, where DM-only achieves a $\mathcal{L}_{DM}$ value of 0.00020. On the other hand, our proposed DBC can effectively optimize both $\mathcal{L}_{BC}$ and $\mathcal{L}_{DM}$, demonstrating the compatibility of the two losses, which justifies the proposed combination of the two losses.

## C    ALLEVIATING MANIFOLD OVERFITTING BY NOISE INJECTION

In section Section 5.4, we show that while our diffusion model loss can enhance the generalization ability of the derived policy, the diffusion models may suffer from manifold overfitting during training and, therefore, need to cooperate with the BC objective. Another branch of machine learning research dealing with overfitting problems is noise injection. As shown in Feng et al. (2021), noise injection regularization has shown promising results that resolve the overfitting problem on image generation tasks. In this section, we evaluate if noise injection can resolve the manifold overfitting directly.

### C.1    MODELING EXPERT DISTRIBUTION

We first verify if noisy injection can help diffusion models capture the expert distribution of the spiral dataset, where the diffusion models fail as shown in Section 5.4. We extensively evaluate diffusion models trained with various levels of noise added to the expert actions. Then, we calculate the average MSE distance between expert actions and the reconstruction of the trained diffusion models, which indicates how well diffusion models capture the expert distribution. We report the result in Table 4.

We observe that applying a noise level of less than 0.02 results in similar MSE distances compared to the result without noise injection (0.0213). The above result indicates that noise injection does not bring an advantage to expert distribution modeling regarding the MSE distance, and the discrepancy between the learned and expert distributions still exists.

### C.2    GUIDE POLICY LEARNING

In order to examine if the noise-injected diffusion model is better guidance for policy, we further investigate the performance of using the learned diffusion models to guide policy learning. Specifically, we train policies to optimize the diffusion model loss $\mathcal{L}_{DM}$ provided by either the diffusion model learning from a noise level of 0 or the diffusion model learning from a noise level of 0.01, dubbed $\pi_{DM\text{-}0.01}$.

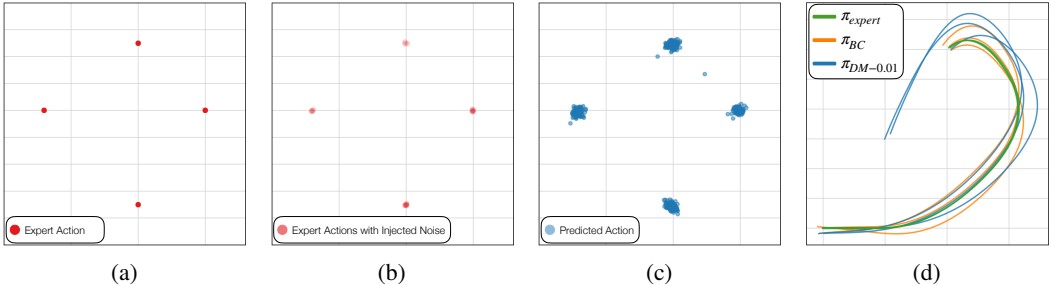

(a)      (b)      (c)      (d)

Figure 6: **Comparing Modeling Conditional Probability and Joint Probability. (a) Expert actions. (b) Expert actions with injected noise. (c) Generated actions by the diffusion model. (d) Rollout trajectories.**

We evaluate the performance of the policies by rolling out each policy and calculating the distance between the end location of the policy and the expert end location. A policy rollout is considered successful if the distance is not greater than 0.1.

In Figure 6, we visualize the expert actions, noise-injected expert actions, generated actions by the diffusion model trained with 0.01 noise level, and the rollout trajectories of the derived policy. The result suggests that the diffusion model learning from expert distribution added with a preferable magnitude noise can better guide policy learning, achieving a success rate of 32%, outperforming the original diffusion model that suffers more from the manifold overfitting with a success rate of 12%. Yet, directly learning to model the conditional probability (i.e., $\pi_{\mathrm{BC}}$) achieves a much higher success rate of 85%. This result verifies the advantage of modeling the conditional probability on this task, which motivates us to incorporate $\mathcal{L}_{\mathrm{BC}}$ in our proposed learning objective instead of solely optimizing $\mathcal{L}_{\mathrm{DM}}$.

## D  COMPARING TO DATA AUGMENTATION

To further explore the usage of diffusion models for improving behavioral cloning, we evaluate a straightforward idea: can diffusion models generate informative samples that enhance the performance of BC?

We leverage the diffusion model learning from an expert dataset to generate state-action pairs as a data augmentation method. Specifically, we use 18525 state-action pairs from the Maze dataset to train a diffusion model and then generate 18525 samples with the trained diffusion model. We combine the real and generated state-action pairs and then learn a BC policy. The policy with data augmentation performs 2.06% better than the one without data augmentation, where the improvement is with a standard deviation, justifying the effectiveness of using the diffusion model as a loss source instead of using it for data augmentation.

## E  ENVIRONMENT & TASK DETAILS

### E.1  MAZE

**Description.** A point-maze agent in a 2D maze learns to navigate from its start location to a goal location by iteratively predicting its x and y acceleration. The 6D states include the agent's two-dimensional current location and velocity, and the goal location. The start and the goal locations are randomized when an episode is initialized.

**Evaluation.** We evaluate the agents with 100 episodes and three random seeds and compare our method with the baselines regarding the average success rate and episode lengths, representing the effectiveness and efficiency of the policy learned by different methods. An episode terminates when the maximum episode length of 400 is reached.

**Expert Dataset.** The expert dataset consists of the 100 demonstrations with 18,525 transitions provided by Lee et al. (2021).

### E.2 FETCHPICK

**Description.** FETCHPICK requires a 7-DoF robot arm to pick up an object from the table and move it to a target location. Following the environment setups of Lee et al. (2021), a 16D state representation consists of the angles of the robot joints, the robot arm poses relative to the object, and goal locations. The first three dimensions of the action indicate the desired relative position at the next time step. The fourth dimension of action specifies the distance between the two fingers of the gripper.

**Evaluation.** We evaluate the agents with 100 episodes and three random seeds and compare our method with the baselines regarding the average success rate and episode lengths. An episode terminates when the agent completes the task or the maximum episode length is reached, which is set to 50 for FETCHPICK.

**Expert Dataset.** The expert dataset of FETCHPICK consists of 303 trajectories ($10k$ transitions) provided by Lee et al. (2021).

### E.3 HANDROTATE

**Description.** HANDROTATE Plappert et al. (2018) requires a 24-DoF Shadow Dexterous Hand to in-hand rotate a block to a target orientation. The 68D state representation consists of the joint angles and velocities of the hand, object poses, and the target rotation. The 20D action indicates the position control of the 20 joints, which can be controlled independently. HANDROTATE is extremely challenging due to its high dimensional state and action spaces. We adapt the experimental setup used in Plappert et al. (2018) and Lee et al. (2021), where the rotation is restricted to the z-axis and the possible initial and target z rotations are set within $[-\frac{\pi}{12}, \frac{\pi}{12}]$ and $[\frac{\pi}{3}, \frac{2\pi}{3}]$, respectively.

**Evaluation.** We evaluate the agents with 100 episodes and three random seeds and compare our method with the baselines regarding the average success rate and episode lengths. An episode terminates when the agent completes the goal or the maximum episode length of 50 is reached.

**Expert Dataset.** To collect expert demonstrations, we train a SAC Haarnoja et al. (2018) policy using dense rewards for $10M$ environment steps. The dense reward given at each time step $t$ is $R(s_t, a_t) = d_t - d_{t+1}$, where $d_t$ and $d_{t+1}$ represent the angles (in radian) between current and the desired block orientations before and after taking the actions. Following the training stage, the SAC expert policy achieves a success rate of 59.48%. Subsequently, we collect 515 successful trajectories ($10k$ transitions) from this policy to form our expert dataset for HANDROTATE.

### E.4 CHEETAH

**Description.** The CHEETAH is a 2D robot with 17 states, indicating the status of each joint. The goal of this task is to exert torque on the joints to control the robot to walk toward x direction.

**Evaluation.** We evaluate each learned policy with 30 episodes and three random seeds and compare our method with the baselines regarding the average returns of episodes. The return of an episode is accumulated from all the time steps of an episode. An episode terminates when the agent is unhealthy (*i.e.*, ill conditions predefined in the environment) or the maximum episode length (1000) is reached.

**Expert Dataset.** The expert dataset consists of 5 trajectories with $5k$ state-action pairs provided by Kostrikov (2018).

### E.5 WALKER

**Description.** WALKER requires an agent to walk toward x-coordinate as fast as possible while maintaining its balance. The 17D state consists of angles of joints, angular velocities of joints, and velocities of the x and z-coordinate of the top. The 6D action specifies the torques to be applied on each joint of the walker avatar.

**Evaluation.** We evaluate each learned policy with 30 episodes and three random seeds and compare our method with the baselines regarding the average returns of episodes. The return of an episode is accumulated from all the time steps of an episode. An episode terminates when the agent is unhealthy (*i.e.*, ill conditions predefined in the environment) or the maximum episode length (1000) is reached.

Table 5: **Model Architectures.** We report the architectures used for all the methods on all the tasks.

| Method | Models | Component | MAZE | FETCHPICK | HANDROTATE | CHEETAH | WALKER | ANTREACH |
|---|---|---|---|---|---|---|---|---|
| BC | Policy $\pi$ | # Layers | 4 | 3 | 3 | 3 | 3 | 3 |
| | | Input Dim. | 6 | 16 | 68 | 17 | 17 | 42 |
| | | Hidden Dim. | 256 | 750 | 1024 | 256 | 256 | 1024 |
| | | Output Dim. | 2 | 4 | 20 | 6 | 6 | 8 |
| Implicit BC | Policy $\pi$ | # Layers | 2 | 2 | 2 | 2 | 2 | 2 |
| | | Input Dim. | 8 | 20 | 88 | 23 | 23 | 50 |
| | | Hidden Dim. | 1024 | 1024 | 512 | 1024 | 1024 | 1200 |
| | | Output Dim. | 1 | 1 | 1 | 1 | 1 | 1 |
| Diffusion Policy | Policy $\pi$ | # Layers | 5 | 5 | 5 | 5 | 5 | 5 |
| | | Input Dim. | 8 | 20 | 88 | 23 | 23 | 42 |
| | | Hidden Dim. | 256 | 1200 | 2048 | 1024 | 1024 | 1200 |
| | | Output Dim. | 2 | 4 | 20 | 6 | 6 | 8 |
| DBC | DM $\phi$ | # Layers | 5 | 5 | 5 | 5 | 5 | 5 |
| | | Input Dim. | 8 | 20 | 88 | 23 | 23 | 50 |
| | | Hidden Dim. | 128 | 1024 | 2048 | 1024 | 1024 | 1024 |
| | | Output Dim. | 8 | 20 | 88 | 23 | 23 | 50 |
| | Policy $\pi$ | # Layers | 4 | 3 | 3 | 3 | 3 | 3 |
| | | Input Dim. | 6 | 16 | 68 | 17 | 17 | 42 |
| | | Hidden Dim. | 256 | 750 | 512 | 1024 | 1024 | 1024 |
| | | Output Dim. | 2 | 4 | 20 | 6 | 6 | 8 |

**Expert Dataset.** The expert dataset consists of $5$ trajectories with $5k$ state-action pairs provided by Kostrikov (2018).

### E.6 ANTREACH

**Description.** The ANTREACH is a 3D robot with four legs, each consisting of two links. The goal of this task is to control the four legs to move the ant toward the goal.

**Evaluation.** We evaluate the agents with $100$ episodes and three random seeds and compare our method with the baselines regarding the average success rate and episode lengths. An episode terminates when the agent completes the goal or the maximum episode length of $60$ is reached.

**Expert Dataset.** We use the demonstrations provided by Lee et al. (2021), which contains 500 trajectories with 25k state-action pairs in this environment.

## F MODEL ARCHITECTURE

This section describes the model architectures used for all the experiments. Section F.1 presents the model architectures of BC, Implicit BC, Diffusion Policy, and our proposed framework DBC. Section F.2 details the model architectures of the EBM, VAE, and GAN used for the experiment comparing different generative models.

### F.1 MODEL ARCHITECTURE OF BC, IMPLICIT BC, DIFFUSION POLICY, AND DBC

We compare our DBC with three baselines (BC, Implicit BC, and Diffusion Policy) on various tasks in Section 5.3. We detail the model architectures for all the methods on all the tasks in Table 5. Note that all the models, the policy of BC, the energy-based model of Implicit BC, the conditional diffusion model of Diffusion Policy, the policy and the diffusion model of DBC, are parameterized by a multilayer perceptron (MLP). We report the implementation details for each method as follows.

**BC.** The non-linear activation function is a hyperbolic tangent for all the BC policies. We experiment with BC policies with more parameters, which tend to severely overfit to expert datasets, resulting in worse performance.

**Implicit BC.** The non-linear activation function is ReLU for all energy-based models of Implicit BC. We empirically find that Implicit BC prefers shallow architectures in our tasks, so we set the number of layers to 2 for the energy-based models.

**Diffusion Policy.** The non-linear activation function is ReLU for all the policies of Diffusion Policy. We empirically find that Diffusion Policy performs better with a deeper architecture. Therefore, we set the number of layers to $5$ for the policy. In most cases, we use a Diffusion Policy with more parameters than the total parameters of DBC consisting of the policy and the diffusion model.

**DBC.** The non-linear activation function is ReLU for the diffusion models and is a hyperbolic tangent for the policies. We apply batch normalization and dropout layers with a $0.2$ ratio for the diffusion models on FETCHPICK.

## F.2    MODEL ARCHITECTURE OF EBM, VAE, AND GAN

We compare different generative models (*i.e.*, EBM, VAE, and GAN) on MAZE in Section 5.5, and we report the model architectures used for the experiment in this section.

**Energy-Based Model.**  An energy-based model (EBM) consists of $5$ linear layers with ReLU activation. The EBM takes a concatenated state-action pair with a dimension of $8$ as input; the output is a $1$-dimensional vector representing the estimated energy values of the state-action pair. The size of the hidden dimensions is $128$.

**Variational Autoencoder.**  The architecture of a variational autoencoder consists of an encoder and a decoder. The inputs of the encoder are a concatenated state-action pair, and the outputs are the predicted mean and variance, which parameterize a Gaussian distribution. We apply the reparameterization trick (Kingma and Welling, 2014), sample features from the predicted Gaussian distribution, and use the decoder to produce the reconstructed state-action pair. The encoder and the decoder both consist of $5$ linear layers with LeakyReLU Xu et al. (2020) activation. The size of the hidden dimensions is $128$. That said, the encoder maps an $8$-dimensional state-action pair to two $128$-dimensional vectors (*i.e.*, mean and variance), and the decoder maps a sampled $128$-dimensional vector back to an $8$-dimensional reconstructed state-action pair.

**Generative Adversarial Network.** The architecture of the generative adversarial network consists of a generator and a discriminator. The generator is the policy model that predicts an action from a given state, whose input dimension is $6$ and output dimension is $2$. On the other hand, the discriminator learns to distinguish the expert state-action pairs $(s, a)$ from the state-action pairs produced by the generator $(s, \hat{a})$. Therefore, the input dimension of the discriminator is $8$, and the output is a scalar representing the probability of the state-action pair being "real." The generator and the discriminator both consist of three linear layers with ReLU activation, and the size of the hidden dimensions is $256$.

# G    TRAINING AND INFERENCE DETAILS

We describe the details of training and performing inference in this section, including computation resources and hyperparameters.

## G.1    COMPUTATION RESOURCE

We conducted all the experiments on the following three workstations:

- M1: ASUS WS880T workstation with an Intel Xeon W-2255 (10C/20T, 19.25M, 4.5GHz) 48-Lane CPU, 64GB memory, an NVIDIA RTX 3080 Ti GPU, and an NVIDIA RTX 3090 Ti GPU

- M2: ASUS WS880T workstation with an Intel Xeon W-2255 (10C/20T, 19.25M, 4.5GHz) 48-Lane CPU, 64GB memory, an NVIDIA RTX 3080 Ti GPU, and an NVIDIA RTX 3090 Ti GPU

- M3: ASUS WS880T workstation with an Intel Xeon W-2255 (10C/20T, 19.25M, 4.5GHz) 48-Lane CPU, 64GB memory, and two NVIDIA RTX 3080 Ti GPUs

Table 6: **Hyperparameters.** This table reports the hyperparameters used for all the methods on all the tasks. Note that our proposed framework (DBC) consists of two learning modules, the diffusion model and the policy, and therefore their hyperparameters are reported separately.

| Method | Hyperparameter | MAZE | FETCHPICK | HANDROTATE | CHEETAH | WALKER | ANTREACH |
|---|---|---|---|---|---|---|---|
| BC | Learning Rate | 5e-4 | 1e-5 | 5e-6 | 1e-4 | 1e-4 | 1e-2 |
| | Batch Size | 128 | 128 | 128 | 128 | 128 | 128 |
| | # Epochs | 2000 | 5000 | 5000 | 1000 | 1000 | 28000 |
| Implicit BC | Learning Rate | 1e-4 | 5e-6 | 1e-4 | 1e-4 | 1e-4 | 1e-4 |
| | Batch Size | 128 | 512 | 128 | 128 | 128 | 128 |
| | # Epochs | 10000 | 15000 | 15000 | 10000 | 10000 | 28000 |
| Diffusion Policy | Learning Rate | 2e-4 | 1e-5 | 1e-5 | 1e-4 | 1e-4 | 1e-5 |
| | Batch Size | 128 | 128 | 128 | 128 | 128 | 128 |
| | # Epochs | 20000 | 15000 | 30000 | 10000 | 10000 | 20000 |
| DBC (Ours) | Diffusion Model Learning rate | 1e-4 | 1e-4 | 3e-5 | 2e-4 | 2e-4 | 2e-4 |
| | Diffusion Model Batch Size | 128 | 128 | 128 | 128 | 128 | 1024 |
| | Diffusion Model # Epochs | 8000 | 10000 | 10000 | 8000 | 8000 | 20000 |
| | Policy Learning Rate | 5e-4 | 1e-5 | 1e-4 | 1e-4 | 1e-4 | 0.008 |
| | Policy Batch Size | 128 | 128 | 128 | 128 | 128 | 128 |
| | Policy # Epochs | 2000 | 5000 | 5000 | 2000 | 2000 | 8000 |
| | $\lambda$ | 30 | 0.1 | 10 | 0.05 | 0.05 | 1 |

## G.2 HYPERPARAMTERS

We report the hyperparameters used for all the methods on all the tasks in Table 6. We use the Adam optimizer Kingma and Ba (2015) for all the methods on all the tasks and use linear learning rate decay for all policy models.

## G.3 INFERENCE DETAILS

This section describes how each method infers an action $\hat{a}$ given a state $s$.

**BC & DBC.** The policy models of BC and DBC can directly predict an action given a state, *i.e.*, $\hat{a} \sim \pi(s)$, and are therefore more efficient during inference as described in Section 5.3.

**Implicit BC.** The energy-based model (EBM) of Implicit BC learns to predict an estimated energy value for a state-action pair during training. To generate a predicted $\hat{a}$ given a state $s$ during inference, it requires a procedure to sample and optimize actions. We follow Florence et al. (2022) and implement a derivative-free optimization algorithm to perform inference.

The algorithm first randomly samples $N_s$ vectors from the action space as candidates. The EBM then produces the estimated energy value of each candidate action and applies the Softmax function on the estimated energy values to produce a $N_s$-dimensional probability. Then, it samples candidate actions according to the above probability and adds noise to them to generate another $N_s$ candidates for the next iteration. The above procedure iterates $N_{iter}$ times. Finally, the action with maximum probability in the last iteration is selected as the predicted action $\hat{a}$. In our experiments, $N_s$ is set to 1000 and $N_{iter}$ is set to 3.

**Diffusion Policy.** Diffusion Policy learns a conditional diffusion model as a policy and produces an action from sampled noise vectors conditioning on the given state during inference. We follow Pearce et al. (2023); Chi et al. (2023) and adopt Denoising Diffusion Probabilistic Models (DDPMs) J Ho (2020) for the diffusion models. Once learned, the diffusion policy $\pi$ can "denoise" a noise sampled from a Gaussian distribution $\mathcal{N}(0, 1)$ given a state $s$ and yield a predicted action $\hat{a}$ using the following equation:

$$a_{n-1} = \frac{1}{\sqrt{\alpha_n}}\left(a_n - \frac{1 - \alpha_n}{\sqrt{1 - \bar{\alpha}_n}}\pi(s, a_n, n)\right) + \sigma_n z, \tag{7}$$

where $\alpha_n$, $\bar{\alpha}_n$, and $\sigma_n$ are schedule parameters, $n$ is the current time step of the reverse diffusion process, and $z \sim \mathcal{N}(0, 1)$ is a random vector. The above denoising process iterates $N$ times to produce a predicted action $a_0$ from a sampled noise $a_N \sim \mathcal{N}(0, 1)$. The number of total diffusion steps $N$ is 100 in our experiment, which is the same for the diffusion model in DBC.

### G.4 COMPARING DIFFERENT GENERATIVE MODELS

Our proposed framework employs a diffusion model (DM) to model the joint probability of expert state-action pairs and utilizes it to guide policy learning. To justify our choice of generative models, we explore using other popular generative models to replace the diffusion model in MAZE. Specifically, we consider energy-based models (EBMs) (Du and Mordatch, 2019; Song and Kingma, 2021), variational autoencoders (VAEs) (Kingma and Welling, 2014), and generative adversarial networks (GANs) Goodfellow et al. (2014). Each generative model learns to model the joint distribution of expert state-action pairs. For fair comparisons, all the policy models learning from learned generative models consists of 3 linear layers with ReLU activation, where the hidden dimension is 256. All the policies are trained for 2000 epochs using the Adam optimizer (Kingma and Ba, 2015), and a linear learning rate decay is applied for EBMs and VAEs.

#### G.4.1 ENERGY-BASED MODEL

**Model Learning.** Energy-based models (EBMs) learn to model the joint distribution of the expert state-action pairs by predicting an estimated energy value for a state-action pair $(s, a)$. The EBM aims to assign low energy value to the real expert state-action pairs while high energy otherwise. Therefore, the predicted energy value can be used to evaluate how well a state-action pair $(s, a)$ fits the distribution of the expert state-action pair distribution.

To train the EBM, we generate $N_{neg}$ random actions as negative samples for each expert state-action pair as proposed in Florence et al. (2022). The objective of the EBM $E_\phi$ is the InfoNCE loss Oord et al. (2018):

$$\mathcal{L}_{\text{InfoNCE}} = \frac{e^{-E_\phi(s,a)}}{e^{-E_\phi(s,a)} + \Sigma_{i=1}^{N_{neg}} e^{-E_\phi(s,\tilde{a}_i)}}, \tag{8}$$

where $(s, a)$ indicates an expert state-action pair, $\tilde{a}_i$ indicates the sampled random action, and $N_{neg}$ is set to 64 in our experiments. The EBM learns to separate the expert state-action pairs from the negative samples by optimizing the above InfoNCE loss.

The EBM is trained for 8000 epochs with the Adam optimizer (Kingma and Ba, 2015), with a batch size of 128 and an initial learning rate of 0.0005. We apply learning rate decay by 0.99 for every 100 epoch.

**Guiding Policy Learning.** To guide a policy $\pi$ to learn, we design an EBM loss $\mathcal{L}_{\text{EBM}} = E_\phi(s, \hat{a})$, where $\hat{a}$ indicates the predicted action produced by the policy. The above EBM loss regularizes the policy to generate actions with low energy values, which encourage the predicted state-action pair $(s, \hat{a})$ to fit the modeled expert state-action pair distribution. The policy learning from this EBM loss $\mathcal{L}_{\text{EBM}}$ achieves a success rate of $49.09\%$ in MAZE as reported in Table 2.

We also experiment with combining this EBM loss $\mathcal{L}_{\text{EBM}}$ with the $\mathcal{L}_{\text{BC}}$ loss. The policy optimizes $\mathcal{L}_{\text{BC}} + \lambda_{\text{EBM}}\mathcal{L}_{\text{EBM}}$, where $\lambda_{\text{EBM}}$ is set to 0.1. Optimizing this combined loss yields a success rate of $80.00\%$ in MAZE as reported in Table 2.

#### G.4.2 VARIATIONAL AUTOENCODER

**Model Learning.** Variational autoencoders (VAEs) model the joint distribution of the expert data by learning to reconstruct expert state-action pairs $(s, a)$. Once the VAE is learned, how well a state-action pair fits the expert distribution can be reflected in the reconstruction loss.

The objective of training a VAE is as follows:

$$\mathcal{L}_{\text{vae}} = ||\hat{x} - x||^2 + D_{\text{KL}}(\mathcal{N}(\mu_x, \sigma_x)||\mathcal{N}(0, 1)), \tag{9}$$

where $x$ is the latent variable, *i.e.*, the concatenated state-action pair $x = [s, a]$, and $\hat{x}$ is the reconstruction of $x$, *i.e.*, the reconstructed state-action pair. The first term is the reconstruction loss, while the second term encourages aligning the data distribution with a normal distribution $\mathcal{N}(0, 1)$, where $\mu_x$ and $\sigma_x$ are the predicted mean and standard deviation given $x$.

The VAE is trained for $100k$ update iterations with the Adam optimizer (Kingma and Ba, 2015), with a batch size of 128 and an initial learning rate of 0.0001. We apply learning rate decay by 0.5 for every $5k$ epoch.

Table 7: **FETCHPICK Generalization Experimental Result.** We report the performance of our proposed framework DBC and the baselines regarding the mean and the standard deviation of the success rate with different levels of noise injected into the initial state and goal locations in FETCHPICK, evaluated over three random seeds.

| Method | Noise Level | | | | |
|---|---|---|---|---|---|
| | 1 | 1.25 | 1.5 | 1.75 | 2 |
| BC | $92.40\% \pm 8.49\%$ | $91.57\% \pm 5.83\%$ | $85.50\% \pm 6.28\%$ | $77.62\% \pm 7.07\%$ | $\mathbf{67.41}\% \pm 8.20\%$ |
| Implicit BC | $83.08\% \pm 3.11\%$ | $69.39\% \pm 7.30\%$ | $51.64\% \pm 4.20\%$ | $36.51\% \pm 4.65\%$ | $23.58\% \pm 2.97\%$ |
| Diffusion Policy | $90.04\% \pm 3.47\%$ | $83.87\% \pm 3.42\%$ | $72.34\% \pm 6.80\%$ | $64.10\% \pm 7.14\%$ | $58.15\% \pm 8.15\%$ |
| DBC (Ours) | $\mathbf{99.53}\% \pm 0.53\%$ | $\mathbf{96.89}\% \pm 1.70\%$ | $\mathbf{91.46}\% \pm 3.30\%$ | $\mathbf{83.30}\% \pm 4.82\%$ | $\mathbf{73.52}\% \pm 6.81\%$ |

**Guiding Policy Learning.** To guide a policy $\pi$ to learn, we design a VAE loss $\mathcal{L}_{\text{VAE}} = max(\mathcal{L}_{\text{vae}}^{\text{agent}} - \mathcal{L}_{\text{vae}}^{\text{expert}}, 0)$, similar to Eq. 5. This loss forces the policy to predict an action, together with the state, that can be well reconstructed with the learned VAE. The policy learning from this VAE loss $\mathcal{L}_{\text{VAE}}$ achieves a success rate of $48.47\%$ in MAZE as reported in Table 2.

We also experiment with combining this VAE loss $\mathcal{L}_{\text{VAE}}$ with the $\mathcal{L}_{\text{BC}}$ loss. The policy optimizes $\mathcal{L}_{\text{BC}} + \lambda_{\text{VAE}}\mathcal{L}_{\text{VAE}}$, where $\lambda_{\text{VAE}}$ is set to 1. Optimizing this combined loss yields a success rate of $82.31\%$ in MAZE as reported in Table 2.

### G.4.3 GENERATIVE ADVERSARIAL NETWORK

**Adversarial Model Learning & Policy Learning.** Generative adversarial networks (GANs) model the joint distribution of expert data with a generator and a discriminator. The generator aims to synthesize a predicted action $\hat{a}$ given a state $s$. On the other hand, the discriminator aims to identify expert the state-action pair $(s, a)$ from the predicted one $(s, \hat{a})$. Therefore, a learned discriminator can evaluate how well a state-action pair fits the expert distribution.

While it is possible to learn a GAN separately and utilize the discriminator to guide policy learning, we let the policy $\pi$ be the generator directly and optimize the policy with the discriminator iteratively. We hypothesize that a learned discriminator may be too selective for a policy training from scratch, so we learn the policy $\pi$ with the discriminator $D$ to improve the policy and the discriminator simultaneously.

The objective of training the discriminator $D$ is as follows:

$$\mathcal{L}_{\text{disc}} = BCE(D(s, a), 1) + BCE(D(s, \hat{a}), 0) = -log(D(s, a)) - log(1 - D(s, \hat{a})), \quad (10)$$

where $\hat{a} = \pi(s)$ is the predicted action, and $BCE$ is the binary cross entropy loss. The binary label $(0, 1)$ indicates whether or not the state-action pair sampled from the expert data. The generator and the discriminator are both updated by Adam optimizers using a $0.00005$ learning rate.

To learn a policy (*i.e.*, generator), we design the following GAN loss:

$$\mathcal{L}_{\text{GAN}} = BCE(D(s, \hat{a}), 1) = -log(D(s, \hat{a})). \quad (11)$$

The above GAN loss guides the policy to generate state-action pairs that fit the joint distribution of the expert data. The policy learning from this GAN loss $\mathcal{L}_{\text{GAN}}$ achieves a success rate of $50.29\%$ in MAZE as reported in Table 2.

We also experiment with combining this GAN loss $\mathcal{L}_{\text{GAN}}$ with the $\mathcal{L}_{\text{BC}}$ loss. The policy optimizes $\mathcal{L}_{\text{BC}} + \lambda_{\text{GAN}}\mathcal{L}_{\text{GAN}}$, where $\lambda_{\text{GAN}}$ is set to $0.2$. Optimizing this combined loss yields a success rate of $71.64\%$ in MAZE as reported in Table 2.

## H GENERALIZATION EXPERIMENTS IN FETCHPICK

This section further investigates the generalization capabilities of the policies learned by our proposed framework and the baselines. To this end, we evaluate the policies by injecting different noise levels to both the initial state and goal location in FETCHPICK. Specifically, we parameterize the noise by scaling the 2D sampling regions for the block and goal locations in both environments. We expect all the methods to perform worse with higher noise levels, while the performance drop of the methods

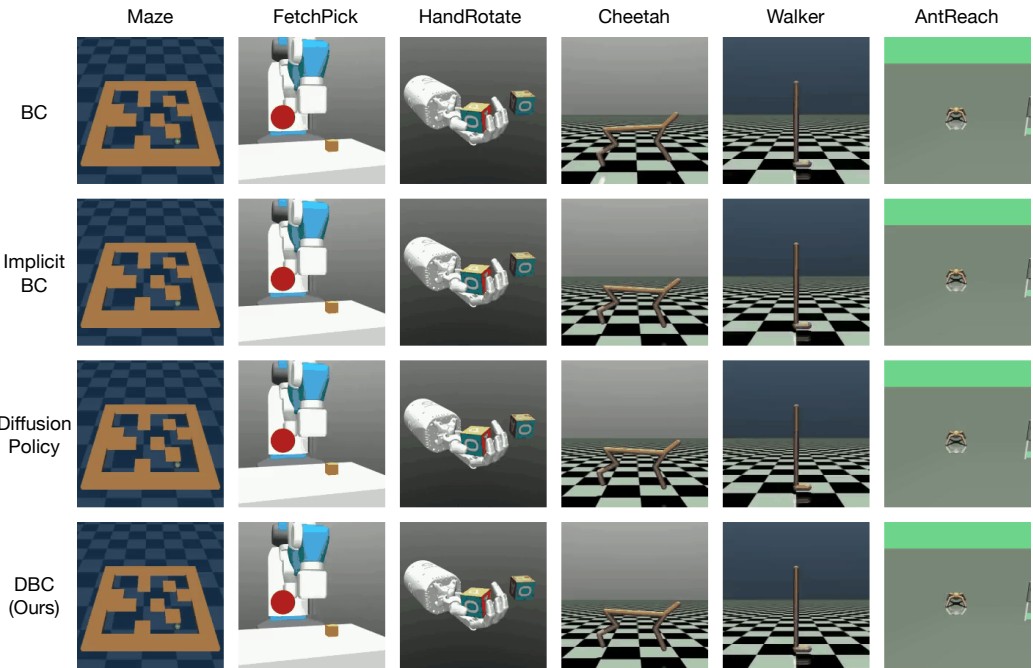

Figure 7: **Qualitative Results.** Rendered videos of the policies learned by our proposed framework and the baselines can be found at `https://sites.google.com/view/diffusion-behavioral-cloning`.

with better generalization ability is less significant. In this experiment, we set the coefficient $\lambda$ of DBC to $0.1$ in FETCHPICK. The results are presented in Table 7 for FETCHPICK.

**Overall Performance.** Our proposed framework DBC consistently outperforms all the baselines with different noise levels, indicating the superiority of DBC when different levels of generalization are required.

**Performance Drop with Increased Noise Level.** In FETCHPICK, DBC experiences a performance drop of $26.1\%$ when the noise level increase from 1 to 2. However, BC and Implicit BC demonstrate a performance drop of $27.0\%$ and $71.6\%$, respectively. Notably, Diffusion Policy initially performs poorly at a noise level of 1 but demonstrates its robustness with a performance drop of only $35.3\%$ when the noise level increases to 2. This demonstrates that our proposed framework not only generalizes better but also exhibits greater robustness to noise compared to the baselines.

## I  QUALITATIVE RESULTS AND ADDITIONAL ANALYSIS

This section provides more detailed analyses of our proposed framework and the baselines. We present the qualitative results in Section I.1. Then, we analyze the learning progress and the episode length of goal-directed tasks during inference in Section **??** and Section **??**, respectively.

### I.1  QUALITATIVE RESULTS

Rendered videos of the policies learned by our proposed framework and the baselines can be found at `https://sites.google.com/view/diffusion-behavioral-cloning`. A screenshot of the rendered videos on the web page is presented in Figure 7.

# J ON THE THEORETICAL MOTIVATION FOR GUIDING POLICY LEARNING WITH DIFFUSION MODEL

This section further elaborates on the technical motivation for leveraging diffusion models for imitation learning. Specifically, we aim to learn a diffusion model to model the joint distribution of expert state-action pairs. Then, we propose to utilize this learned diffusion model to augment a BC policy that aims to imitate expert behaviors.

We consider the distribution of expert state-action pairs as the real data distribution $q_x$ in learning a diffusion model. Following this setup, $x_0$ represents an original expert state-action pair $(s, a)$ and $q(x_n|x_{n-1})$ represents the forward diffusion process, which gradually adds Gaussian noise to the data in each timestep $n = 1, ..., N$ until $x_N$ becomes an isotropic gaussian distribution. On the other hand, the reverse diffusion process is defined as $\phi(x_{n-1}|x_n) := \mathcal{N}(x_{n-1}; \mu_\theta(x_n, n), \Sigma_\theta(x_n, n))$, where $\theta$ denotes the learnable parameters of the diffusion model $\phi$, as illustrated in Figure 1.

Our key idea is to use the proposed diffusion model loss $\mathcal{L}_{DM}$ in Eq. 5 as an estimate of how well a predicted state-action pair $(s, \hat{a})$ fits the expert state-action pair distribution, as described in Section 4.2.2. In the following derivation, we will show that by optimizing this diffusion model loss $\mathcal{L}_{DM}$, we maximize the lower bound of the agent data's probability under the derived expert distribution and hence bring the agent policy $\pi$ closer to the expert policy $\pi^E$, which is the goal of imitation learning.

As depicted in Luo (2022), one can conceptualize diffusion models, including DDPM (J Ho, 2020) adopted in this work, as a hierarchical variational autoencoder (Kingma and Welling, 2014), which maximizes the likelihood $p(x)$ of observed data points $x$. Therefore, similar to hierarchical variational autoencoders, diffusion models can optimize the Evidence Lower Bound (ELBO) by minimizing the KL divergence $D_{KL}(q(x_{n-1}|x_n, x_0)||\phi(x_{n-1}|x_n))$. Consequently, this can be viewed as minimizing the KL divergence to fit the distribution of the predicted state-action pairs $(s, \hat{a})$ to the distribution of expert state-action pairs.

According to Bayes' theorem and the properties of Markov chains, the forward diffusion process $q(x_{n-1}|x_n, x_0)$ follows:

$$q(x_{n-1}|x_n, x_0) \sim \mathcal{N}(x_{n-1}; \underbrace{\frac{\sqrt{\alpha_n}(1 - \bar{\alpha}_{n-1})x_n + \sqrt{\bar{\alpha}_{n-1}}(1 - \alpha_n)x_0}{1 - \bar{\alpha}_n}}_{\mu_q(x_n, x_0)},$$

$$\underbrace{\frac{(1 - \alpha_n)(1 - \bar{\alpha}_{n-1})}{1 - \bar{\alpha}_n}}_{\Sigma_q(n)}).$$

The variation term $\Sigma_q(n)$ in the above equation can be written as $\sigma_q^2(n)I$, where $\sigma_q^2(n) = \frac{(1 - \alpha_n)(1 - \bar{\alpha}_{n-1})}{1 - \bar{\alpha}_n}$. Therefore, minimizing the KL divergence is equivalent to minimizing the gap between the mean values of the two distributions:

$$\arg\min_\theta D_{KL}(q(x_{n-1}|x_n, x_0)||\phi(x_{n-1}|x_n))$$

$$= \arg\min_\theta D_{KL}(\mathcal{N}(x_{n-1}; \mu_q, \Sigma_q(n))||\mathcal{N}(x_{n-1}; \mu_\theta, \Sigma_q(n)))$$

$$= \arg\min_\theta \frac{1}{2\sigma_q^2(n)}[||\mu_\theta - \mu_q||_2^2],$$

where $\mu_q$ represents the denoising transition mean and $\mu_\theta$ represents the approximated denoising transition mean by the model.

Different implementations adopt different forms to model $\mu_\theta$. Specifically, for DDPMs adopted in this work, the true denoising transition mean $\mu_q(x_n, x_0)$ derived above can be rewritten as:

$$\mu_q(x_n, x_0) = \frac{1}{\sqrt{\alpha_n}}(x_n - \frac{1 - \alpha_n}{\sqrt{1 - \bar{\alpha}_n}}\epsilon_0),$$

which is referenced from Eq. 11 in J Ho (2020). Hence, we can set our approximate denoising transition mean $\mu_\theta$ in the same form as the true denoising transition mean:

$$\mu_\theta(x_n, n) = \frac{1}{\sqrt{\alpha_n}}(x_n - \frac{1 - \alpha_n}{\sqrt{1 - \bar{\alpha}_n}}\hat{\epsilon}_\theta(x_n, n)), \tag{12}$$

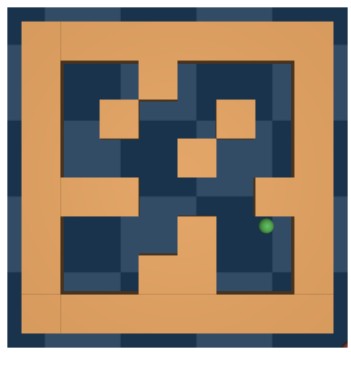
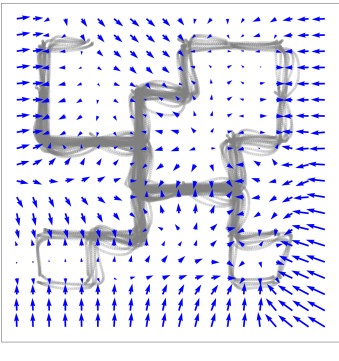

(a) **Maze Layout**                    (b) **Learned Gradient Field**

Figure 8: **Visualized Gradient Field. (a) Maze Layout**: The layout of the medium maze used for MAZE. **(b) Learned Gradient Field**: We visualize the MAZE expert demonstration as a distribution of points by their first two dimensions in gray. The points that cluster densely have a high probability, and vice versa. Once a diffusion model is well-trained, it can move randomly sampled points to the area with high probability by predicting gradients (blue arrows). Accordingly, the estimate $p(s, a)$ of joint distribution modeling can serve as guidance for policy learning, as proposed in this work.

as illustrated in Popov et al. (2022). Song et al. (2021) further show that the entire diffusion model formulation can be revised to view continuous stochastic differential equations (SDEs) as a forward diffusion. It points out that the reverse process is also an SDE, which can be computed by estimating a score function $\nabla_x \log p_t(x)$ at each denoising time step. The idea of representing a distribution by modeling its score function is introduced in Song and Ermon (2019). The fundamental concept is to model the gradient of the log probability density function $\nabla_x \log p_t(x)$, a quantity commonly referred to as the (Stein) score function. Such score-based models are not required to have a tractable normalizing constant and can be directly acquired through score matching. The measure of this score function determines the optimal path to take in the space of the data distribution to maximize the log probability under the derived real distribution.

As shown in Figure 8b, we visualized the learned gradient field of a diffusion model, which learns to model the expert state-action pairs in MAZE. Once trained, this diffusion model can guide a policy with predicted gradients (blue arrows) to move to areas with high probability, as proposed in our work.

Essentially, by moving in the opposite direction of the source noise, which is added to a data point $x_t$ to corrupt it, the data point is "denoised"; hence the log probability is maximized. This is supported by the fact that modeling the score function is the same as modeling the negative of the source noise. This perspective of the diffusion model is dubbed diffusion SDE. Moreover, Popov et al. (2022) prove that Eq. 12 is diffusion SDE's maximum likelihood SDE solver. Hence, the corresponding divergence optimization problem can be rewritten as:

$$\arg\min_{\theta} D_{KL}(q(x_{n-1}|x_n, x_0)||\phi(x_{n-1}|x_n))$$

$$= \arg\min_{\theta} \frac{1}{2\sigma_q^2(n)} \frac{(1-\alpha_n)^2}{(1-\bar{\alpha}_n)\alpha_n} [||\hat{\epsilon}_{\theta}(x_n, n) - \epsilon_0||_2^2],$$

where $\epsilon_{\theta}$ is a function approximator aim to predict $\epsilon$ from $x$. As the coefficients can be omitted during optimization, we yield the learning objective $\mathcal{L}_{\text{diff}}$ as stated in in Eq. 2:

$$\mathcal{L}_{\text{diff}}(s, a, \phi) = \mathbb{E}_{n \sim N, (s,a) \sim D}\{||\hat{\epsilon}(s, a, n) - \epsilon(n)||^2\} = \mathbb{E}_{n \sim N, (s,a) \sim D}\{||\phi(s, a, \epsilon(n)) - \epsilon(n)||^2\}.$$

The above derivation motivates our proposed framework that augments a BC policy by using the diffusion model to provide guidance that captures the joint probability of expert state-action pairs. Based on the above derivation, minimizing the proposed diffusion model loss (*i.e.*, learning to denoise) is equivalent to finding the optimal path to take in the data space to maximize the log probability. To be more accurate, when the learner policy predicts an action that obtains a lower $\mathcal{L}_{\text{diff}}$, it means that the predicted action $\hat{a}$, together with the given state $s$, fits better with the expert distribution.

Accordingly, by minimizing our proposed diffusion loss, the policy is encouraged to imitate the expert policy. To further alleviate the impact of rarely-seen state-action pairs $(s, a)$, we propose to compute the above diffusion loss for both expert data $(s, a)$ and predicted data $(s, \hat{a})$ and yield $\mathcal{L}_{\text{diff}}^{\text{expert}}$ and $\mathcal{L}_{\text{diff}}^{\text{agent}}$, respectively. Therefore, we propose to augment BC with this objective: $\mathcal{L}_{\text{DM}} = \mathbb{E}_{(s,a)\sim D, \hat{a}\sim\pi(s)}\{max(\mathcal{L}_{\text{diff}}^{\text{agent}} - \mathcal{L}_{\text{diff}}^{\text{expert}}, 0)\}$.

## K    LIMITATIONS

This section discusses the limitations of our proposed framework.

- Since this work aims to learn from demonstrations without interacting with environments, our proposed framework in its current form is only designed to learn from expert trajectories and cannot learn from trajectories produced by the learner policy. Extending our method to incorporate agent data can potentially allow for improvement when interacting environments are possible, which is left for future work.
- The key insight of our work is to allow the learner policy to benefit from both modeling the conditional and joint probability of expert state-action distributions. To this end, we propose to optimize both the BC loss and the proposed diffusion model loss. To balance the importance of the two losses, we introduce a coefficient $\lambda$ as an additional hyperparameter. While the ablation study conducted in MAZE shows that the performance of our proposed framework is robust to $\lambda$, this can potentially increase the difficulty of searching for optimal hyperparameters when applying our proposed framework to a new application.

## L    BROADER IMPACTS

This work proposes Diffusion Model-Augmented Behavioral Cloning, a novel imitation learning framework that aims to increase the ability of autonomous learning agents (*e.g.*, robots, game AI agents) to acquire skills by imitating demonstrations provided by experts (*e.g.*, humans). However, it is crucial to acknowledge that our proposed framework, by design, inherits any biases exhibited by the expert demonstrators. These biases can manifest as sub-optimal, unsafe, or even discriminatory behaviors. To address this concern, ongoing research endeavors to mitigate bias and promote fairness in machine learning hold promise in alleviating these issues. Moreover, research works that enhance learning agents' ability to imitate experts, such as this work, can pose a threat to job security. Nevertheless, in sum, we firmly believe that our proposed framework can offer tremendous advantages in terms of enhancing the quality of human life and automating laborious, arduous, or perilous tasks that pose risks to humans, which far outweigh the challenges and potential issues.