# OpenReview forum: "Diffusion Model-Augmented Behavioral Cloning"
_ICLR.cc/2024/Conference — ICLR 2024 Conference Withdrawn Submission_

### Official Review · Reviewer_Mkdd · 2023-10-30

**Soundness:** 3 good
**Presentation:** 3 good
**Contribution:** 3 good
**Rating:** 6
**Confidence:** 4

**Summary:**

This paper addresses the imitation learning problem and introduces a novel approach that combines the power of a diffusion model for predicting the joint distribution p(s, a) with standard behavior cloning to estimate the conditional distribution p(a | s). The proposed method is evaluated through a series of experiments spanning diverse tasks, including navigation, robot manipulation, and locomotion.

**Strengths:**

1. Compared to the baseline approach of the Diffusion Policy, the proposed method does not directly utilize the diffusion model to generate final actions. Instead, the diffusion model is used as an implicit evaluator or data augmenter to provide additional learning signals to the policy network. The proposed method introduces a novel perspective in the field of imitation learning by providing the learning agent with additional signals using the diffusion model. The paper could emphasize more on this new perspective.

2. The study carries out experiments across a wide range of tasks, including navigation, robotic manipulation, and locomotion. The methodology and results presented in this research could have significant implications for future work in this arena.

**Weaknesses:**

1. The performance improvement is not significant. Compared with vanilla BC, the proposed method does not have a significant improvement on CHEETAH WALKER ANTREACH HANDROTATE.  Only has a slight improvement on MAZE (about 3%) FETCHPICK (about 5%).  Considering conduct experiments on tasks that BC performs badly and DM performs well to show DBC get the merits of both BC and DM.

**Questions:**

1. Interesting to know where the transformer structure sits in Table2:GenerativeModels. E.g. use transformers to generate next step robot states.

2. Interesting to know how much Equation 5 brings to much compared to Equation 3 alone.

---

### Official Review · Reviewer_dYN2 · 2023-11-01

**Soundness:** 2 fair
**Presentation:** 3 good
**Contribution:** 2 fair
**Rating:** 3
**Confidence:** 4

**Summary:**

In this study, the authors propose an approach to enhance the generalizability of behavioral cloning (BC) agents when faced with states that lie outside the training data distribution. They introduce an additional diffusion model loss alongside the original BC loss to train the agent. Experimental results demonstrate that the proposed method outperforms baseline methods.

**Strengths:**

1. The training procedure for the proposed method appears to be straightforward, relying solely on supervised learning.
2. The proposed method exhibits strong performance across multiple benchmark scenarios.

**Weaknesses:**

See Questions

**Questions:**

1. Theoretically, the agent can output the expert’s action after training with only BC loss. When reach optimal, these two state-action pairs $(s^e,\pi(s^e))$,$(s^e,a^e)$ are the same. Since $(s^e, a^e)$ constitutes the training data for the diffusion model, $(s^e, \pi(s^e))$ should theoretically represent the optimal point on the diffusion model loss surface, with a gradient of zero. It remains unclear how the agent can gain an advantage from the additional diffusion model loss in this scenario.
2. It would be valuable to explore the performance of BC when combined with regularization techniques such as L1 or L2 weight penalties.

---

### Official Review · Reviewer_bGrV · 2023-11-03

**Soundness:** 2 fair
**Presentation:** 2 fair
**Contribution:** 1 poor
**Rating:** 3
**Confidence:** 5

**Summary:**

This paper uses behavior cloning (BC) to train task policies. But instead of solely learning the task policy p(a|s) they also learn the joint probability distribution p(s, a), which can be seen as a transition dynamics of the provided expert distributions. The latter is learned using a diffusion model from which (s, a) pairs can be generated. Experiments are performed on simple domains and some slightly more challenging manipulation tasks.

**Strengths:**

The paper is easy to read and understand. Diffusion models have shown to be quite effective for many control tasks. Thus, understanding and developing new improved techniques in using them for control is an important research problem.

**Weaknesses:**

The proposed approach aims to first model p(s, a) pairs that is it trains a generative model to predict both the state as well as the actions. This is definitely less than ideal for high dimensional state inputs such as images. Second it is unclear how well this diffusion model will work for multi-task policies or even policies in the same environments but where state inputs can have wide distributions. The examples being considered in the paper only focus on very simple toy envs (cheetah, walker) or objects with a single target object (cube for robot manipulation).

**Results:** For the complexity of the proposed approach the environments used for evaluation are relatively toy. Moreover, the proposed approach does not even show any significant gains compared to existing approaches. This is especially true considering that BC policies probably have very few parameters while the used diffusion models will have much larger number of parameters and a more complex training strategy.

**Novelty and related work:** The paper proposes to model (s, a) pairs using a diffusion model. However, for deterministic environments (all mujoco envs considered here), this is similar to modelling (s, a, s’) pairs. Howevever, many prior works focus on modeling both the transition dynamics (s, a, s’) and the policy together. Infact, one of the original papers (Janner et al.), which proposed using diffusion model for control tasks motivated diffusion models with a very similar motivation. Specifically, Janner et al. learn both the transition dynamics as well as the underlying policy of the task. Unfortunately, Janner et al. paper is neither cited nor discussed. One difference between Janner et al. and the current paper is that the current paper is only predicing per state (s, a) pairs instead of an entire trajectory. However, I don’t see that as a huge difference, infact I think modeling the temporal dynamics with diffusion models is probably the right thing to do (instead of per-trajectory options — this is what diffusion policy (Chi et al.) also does). The current paper is light on specific details, but it seems the current model only uses 1-step (s, a) for training diffusion model instead of trajectories (maybe the authors can correct me if I am wrong).

**Distillation:** The paper can also be considered from a distillation perspective where a diffusion model is learned and then distilled into an MLP policy using the trajectories generated by it. Maybe works that consider distillation with diffusion models should also be considered. Further, some recent work in multi-agent control (MADiff) with diffusion model also has a similar perspective where a diffusion model is used as a generative model to generate data for training. Maybe these works should also be cited.

Meng et al. On Distillation of Guided Diffusion Models

Zhu et al. MADiff: Offline Multi-agent Learning with Diffusion Models

Overall, I am unclear if there is anything new that a reader is learning from this paper. The main idea of the paper has been shown previously in slightly different setting but in a much more broader context.

**Questions:**

Please see above

---

### Official Review · Reviewer_8bWB · 2023-11-15

**Soundness:** 1 poor
**Presentation:** 2 fair
**Contribution:** 2 fair
**Rating:** 3
**Confidence:** 4

**Summary:**

This paper introduces an approach to enhance behavior cloning through the learning of state-action joint distributions. In addition to an imitation learning model trained to mimic expert actions conditioned on state observations, the authors propose to use a diffusion model to maximize the log-likelihood of state and action pairs by learning a joint distribution over the joint space. The authors verify model performance usign a variety of generative model choices, and compare against three imitation learning methods.

**Strengths:**

1) The paper is easy to read and well-structured. The authors present a novel diffusion model for learning a joint distribution over state-action pairs which is presented clearly.
2) The 6 tasks chosen for evaluation are challenging and results show improvements over the considered baselines.

**Weaknesses:**

1) The motivation for learning a joint distribution over states and actions is unclear. The authors aim to address this in Section 3.1.2 but I am not convinced that the cited works clearly support their claims. For instance, the authors mention that “[IBC] (Florence et al., 2022) propose to model the joint probability p(s,a) of expert state-action pairs using an energy-based model,” which is not ture. While the energy function formulation in IBC takes both state and action as input, it models a conditional distribution over the action space and not a joint distribution.
2) I also think that learning the conditional and joint distributions separately introduces redundancy to the model, which is not addressed in the paper.
3) Explanation for the choice of baselines (Section 5.2) seems very undermotivated. I think there was avenue here to explain the design choices in these methods and why they make a diverse set of baselines.

**Questions:**

Please refer to the section above.

**Details Of Ethics Concerns:**

No concerns.

---

### Author Response · Authors · 2023-11-15
**Paper Withdrawal**

After careful consideration, we have decided to withdraw this submission. We sincerely thank all the reviewers and the AC for putting so much effort into helping us to improve this submission. We will work on addressing the concerns raised by the reviewers.